# Context-Dependent Effects Explain Divergent Prognostic Roles of Tregs in Cancer

**DOI:** 10.3390/cancers14122991

**Published:** 2022-06-17

**Authors:** Elise Amblard, Vassili Soumelis

**Affiliations:** 1Université Paris Cité, Inserm U976, 75006 Paris, France; vassili.soumelis@aphp.fr; 2Institut de Recherche Saint-Louis, 75010 Paris, France; 3Assistance Publique-Hôpitaux de Paris (AP-HP), Hôpital Saint-Louis, Laboratoire d’Immunologie, 75010 Paris, France

**Keywords:** regulatory T cells, cancer prognosis, context dependency

## Abstract

**Simple Summary:**

Immune cells play an important role in cancer, with regard to classification, diagnostic or prognostic matters. In particular, we focused on the prognostic value of Tregs in this meta-analysis. We took into account the local context and their heterogeneity in order to solve their apparent ambiguous role. We used three proxies to recapitulate the complexity of the context: the neighboring cell, the tissue and the quantification method; and we carefully dissected the regulatory population into existing subsets. We showed that CD45RO${}^{+}$ Tregs had a reproducible negative prognostic value across all five cancer types studied (breast, colorectal, gastric, lung and ovarian). It suggests that Tregs from an homogeneous context have a consistent prognostic role across cancer types.

**Abstract:**

Assessing cancer prognosis is a challenging task, given the heterogeneity of the disease. Multiple features (clinical, environmental, genetic) have been used for such assessments. The tumor immune microenvironment (TIME) is a key feature, and describing the impact of its many components on cancer prognosis is an active field of research. The complexity of the tumor microenvironment context makes it difficult to use the TIME to assess prognosis, as demonstrated by the example of regulatory T cells (Tregs). The effect of Tregs on prognosis is ambiguous, with different studies considering them to be negative, positive or neutral. We focused on five different cancer types (breast, colorectal, gastric, lung and ovarian). We clarified the definition of Tregs and their utility for assessing cancer prognosis by taking the context into account via the following parameters: the Treg subset, the anatomical location of these cells, and the neighboring cells. With a meta-analysis on these three parameters, we were able to clarify the prognostic role of Tregs. We found that CD45RO${}^{+}$ Tregs had a reproducible negative effect on prognosis across cancer types, and we gained insight into the contributions of the anatomical location of Tregs and of their neighboring cells on their prognostic value. Our results suggest that Tregs play a similar prognostic role in all cancer types. We also establish guidelines for improving the design of future studies addressing the pathophysiological role of Tregs in cancer.

## 1. Introduction

In the last decades, much cancer research has focused on immunity, with the aim of disentangling the links spanning the tumor immune microenvironment (TIME) and understanding why immune cells fail to eradicate malignant tumors. In particular, the field of immunotherapy, which aims to boost the destruction of cancer cells by the host immune system, is growing rapidly. There is evidence suggesting that the TIME is a key predictor of the clinical course of the disease in humans, from tumorigenesis [1,2] to global prognosis [3,4], risk of metastasis [5,6], and response to treatment [7]. Some TIME features, such as Immunoscore, are used for tumor classification in clinical practice and laboratory studies [8,9], for many different cancer types, including melanoma, prostate, breast, lung, and colorectal tumors.

Many reviews have summarized the prognostic role of immune cells from the TIME or the peripheral blood in various types of cancer [10,11,12]. Tregs appear to be highly versatile and to have the most ambiguous prognostic role of the immune populations studied. Depending on the cancer type and study considered, Tregs may be considered to be associated with a good or a poor prognosis, or to have no impact on prognosis.

The role of Tregs is highly complex. These cells are involved in maintaining peripheral tolerance and suppressing auto-immunity and inflammation, but they may also prevent antitumor immunity. These different functions are explained by the fact that the Treg population is highly heterogeneous.

Tregs have several different origins, and this is one source of their heterogeneity. Some Tregs, the thymic or natural Tregs (nTregs), are produced in the thymus and released into the peripheral blood [13]. The remaining Tregs develop in response to the stimulation of naive peripheral CD4${}^{+}$ T cells and are described as peripheral (in vivo) or induced (in vitro) Tregs (pTregs and iTregs, respectively). In humans, nTregs display demethylation of the Treg-specific demethylated region (TSDR) of the Foxp3 promoter, leading to the stable expression of FOXP3. By contrast, freshly differentiated iTregs and pTregs display methylation of the TSDR [14], leading to a phenotype more plastic than that of nTregs and a more volatile commitment to the regulatory lineage, although chronically stimulated iTregs also display TSDR demethylation. It has been suggested that, in the context of cancer, most tumor-infiltrating Tregs are pTregs, diverted to a regulatory phenotype by the local microenvironment [15].

A second source of heterogeneity is the diversity of suppressing mechanisms: Tregs may exert suppression on antigen-presenting cells (APCs) or other T cells in a contact-dependent or -independent manner. Tregs targeting APCs can result in poor antigen presentation by the latter. This modulation of APC phenotype can lead to CD4${}^{+}$ T cells developing a regulatory phenotype and an impaired response of CD8${}^{+}$ T cells. Tregs can also interact with effector T cells directly, via various mechanisms. They can kill T cells by releasing perforins or granzymes [16], impair their functions by releasing inhibitory cytokines, such as IL-10, IL-35 [17], and TGF-β [18], or disturb their metabolism. Each of these mechanisms is elicited by context-specific cues, triggering a myriad of modes of action for regulation, further expanding the diversity of Tregs.

Tregs can also be classified into subsets with different suppression potentials, potentially targeting specific cell populations, such as Th1, Th2, Th17 or Th22 cells.

Tregs play a key role in cancer, through the modulation of host response to the tumor and to treatment: they may be crucial targets for treatment or jeopardize or improve the response to treatments targeting other cell types. However, as indicated in reviews on the subject, their contribution to the prognosis of human cancers remains unclear, as it appears to depend on both the cancer type and the study considered. We hypothesized that this ambiguity can be related to Treg heterogeneity [19,20,21] and context. To the best of our knowledge, no previous study has ever focused on the effect of context, most of them having rather focused on the tumor itself: tumor site, type, or stage, for example, [22].

In this meta-analysis, we aimed to clarify the contribution of Tregs to the prognosis of human cancers, by adopting a context-dependent approach to the problem. We used various parameters relating to Treg context to investigate the role of these cells in determining prognosis. Where possible, we extracted the following parameters: (i) the markers used for Treg definition; (ii) the anatomic location; (iii) the technique used to identify Tregs; and (iv) the cells from the same local environment as the Tregs (named neighboring cells).

We first tried to identify a consensus, in cancer studies, as to the best markers of Tregs to use. We then investigated the link between Tregs, their context and cancer prognosis, with the aim of shedding light on the prognostic role of Tregs.

## 2. Materials & Methods

### 2.1. Articles Selection

We searched PubMed for all articles (as of 2020) related to our topic, with the search words “Humans” [Mesh] AND “T-Lymphocytes, Regulatory” [Mesh] AND (“Treg” [Title/Abstract] OR “Tregs” [Title/Abstract] OR “regulatory T” [Title/Abstract]), adding the cancer in which we were interested—breast, colorectal, gastric, lung and ovarian cancers—as a MeSH term. We added a second filter, narrowing the selection to articles published in a journal with an impact factor above 2 (we used the 2-years impact factor of 2019). We identified 81 articles for breast cancer, 76 for colorectal cancer, 47 for gastric cancer, 87 for lung cancer and 50 for ovarian cancer. Finally, we also added the following exclusion criteria: focus on cells other than human Tregs (mouse Tregs, regulatory B cells, CD8${}^{+}$ Tregs), focus outside the primary tumor (metastasis, relapsed cancer, tumor lines, in vitro systems), patients treated with immunotherapy or in a context of dysimmunity, missing information (markers used, anatomic location, number of patients), review articles. In particular, we excluded studies with patients treated by immunotherapy as we were interested in studying an unperturbed TIME. We have, however, included studies concluding on the prognostic role of Tregs with the whole cohort (PMIDS 17135638, 22842982, 23075422, 22879926), or only part of the cohort (PMIDS 21521526, 27494875, 19064967, 24675384, 31681276, 21792941) having received chemotherapy, radiotherapy or hormonal therapy. We ended up with a total of 23 articles on breast cancer, 28 on colorectal cancer, 24 on gastric cancer, 35 on lung cancer and 20 on ovarian cancer, and the selection process has been conducted by one of the authors.

We used the PRISMA guidelines to ensure the quality of the article selection (http://prisma-statement.org/PRISMAStatement/Checklist as of 1 June 2022) and were consistent with applicable items, namely the ones regarding the introduction and methods (items 3 to 10, 16, 17, and 23 to 27). This review was not registered in PROSPERO.

### 2.2. Features Retrieval

We studied three context-related parameters: the Treg population investigated (defined by specific markers), the anatomic location, and the Treg quantification method. All studies combined, this analysis included 3996 patients with breast cancer, 6040 with colorectal cancer, 2015 with gastric cancer, 2359 with lung cancer, and 1754 with ovarian cancer (see references of all articles included in Appendix A.

For each article, we extracted the following features: the PMID, the number of patients included in the study, if the cohort was treated or not, the localisation of the samples studied (classified into intra- or peritumoral, nest, stroma, tumor, blood, juxtatumor, lymph node or tertiary lymphoid structure), the markers used to define the Treg (sub)population, the detection method (FACS or IHC), if the Tregs were tested for functionality in a suppression assay, the method of quantification (absolute or ratio), the neighboring cells measured along Tregs, and the prognosis. The feature extraction was performed by one author.

Regarding the prognosis, we sorted the articles into three categories: good or poor prognosis (Tregs were associated positively, respectively negatively, to prognosis) or neutral prognosis (the authors found no significant association between Tregs and prognosis). We considered simultaneously all events related to prognosis: relapse-free survival, overall survival, disease-free survival.

### 2.3. Regulatory T Cells Definition

To get an overview of the regulatory markers, we pooled all articles from the different cancer types, and retained only articles stating to be studying the whole Treg population. We listed from the resulting 112 articles the markers used and used hierarchical clustering to examine co-occurrences in FACS studies.

### 2.4. Evaluation of the Degree of Agreement

We used normalized Shannon entropy, and Fleiss’ kappa to evaluate the degree of consensus. Fleiss’ kappa is calculated as follows:(1)$$\kappa =\frac{\sum_{i}N_{i}\left(N_{i}-1\right)}{\sum_{i}N_{i}\left(\sum_{i}N_{i}-1\right)},$$
where $N_{i}$ is the number of raters opting for choice *i*. In our case, the choice was either −1 for a poor prognosis, 0 for a neutral prognosis or 1 for a good prognosis, and the number of raters corresponds to the number of patients. Shannon entropy is calculated as follows:(2)$$1-\mathrm{SE}=1+\frac{1}{\log_{10}\sum_{j}1}\sum_{i}\frac{N_{i}}{\sum_{j}N_{j}}\log_{10}\frac{N_{i}}{\sum_{j}N_{j}}.$$

## 3. Results

### 3.1. The Definition of Tregs Is Fuzzy in Human Cancer Literature

Our review of the human Treg literature revealed an absence of consensus on the markers used to identify Tregs. In mice, FOXP3 is a specific marker of Tregs, whereas in humans, FOXP3 is also transiently expressed by effector cells or ex-Tregs [23,24,25], and is not expressed at all [26,27] or only at low levels [28] by certain Treg subsets. Historically, the definition of Tregs was functional. These cells were first described as T cells that regulated immunity by exerting suppression, a definition that could even include CD8${}^{+}$ Tregs [29,30]. However, this functional description encompasses heterogeneous subpopulations: there is both ontogenic and phenotypic heterogeneity, linked to considerable functional diversity [28].

We focused our analysis on human studies of five cancer types, to ensure that the analysis spanned the entire spectrum of Treg prognostic values. Tregs from breast and lung cancers (non-small cell lung cancer, NSCLC) are negatively associated with clinical outcome, whereas the association is positive for gastric and colorectal cancers, and there seems to be no benefit from using Tregs for prognostic applications in ovarian cancer [11]. We decided to focus exclusively on CD4${}^{+}$ Tregs, because too little had been published on the role of CD8${}^{+}$ Tregs in cancer for a reliable analysis: we found only three articles on CD8${}^{+}$ Tregs [31,32,33] and one on CD4${}^{+}$CD8${}^{+}$ Tregs [34] among the 341 articles we analyzed (Figure 1).

We evaluated the degree of consensus concerning the definition of Tregs, by pooling all articles for the five cancer types considered, a total of 130 publications (23 for breast, 28 for colorectal, 24 for gastric, 35 for lung and 20 for ovarian cancers. See Methods for article selection, Appendix A and Figure 1).

With the aim of highlighting markers of the regulatory population, we removed the articles focusing exclusively on certain subsets, leaving us with a total of N = 112 articles studying the whole Treg population. We distinguished between two methods of Treg detection: fluorescence-activated cell sorting (FACS) in 60% of the publications and immunohistochemistry (IHC). We also noted whether these methods were used in conjunction with a working suppression assay (Figure 2a). This was the case for 41% of FACS studies and 5% of IHC studies. CD4 and CD25 were routinely used as markers in FACS studies, together with CD127 and the transcription factor FOXP3. The use of a combination of CD4 and CD25 was also very common. The use of other markers, such as CXCR5 and CD69, was more anecdotal, and these markers were always used in combination with one or more of the classical markers, CD4, CD25 and FOXP3. Some studies applied thresholds for the expression of certain markers: e.g., 29% of the FACS articles in which Tregs were detected with CD25 used CD25^high^ rather than CD25${}^{+}$ (71% of the articles), and 25% used CD127^low^ rather than CD127${}^{-}$ (75% of the articles). For IHC, the ubiquitous marker was FOXP3, with CD4 and CD25 used more sparingly. The only study using CD4 and CD25 rather than FOXP3 to delineate Tregs did so because the aim was to investigate FOXP3 not only in Tregs, but also in cancer cells [35]. The use of multiple markers was rare for IHC, due to technical challenges.

We studied the individual cancer types one by one. It was not possible to collect equal numbers of articles for each cancer type (Figure 2b, left panel). We first determined, for each cancer type, whether there were equal proportions of articles using FACS or IHC for Treg detection. This was the case for all cancers except lung cancer, for which a higher proportion of articles were based on blood samples, which therefore used only FACS (Figure 2b, right panel). Across cancer types and for IHC, there was a general consensus in favor of FOXP3 staining. The diversity of markers was broader for FACS, with some studies using large numbers of markers, particularly those concerning lung and colorectal cancers, for which the mean number of markers used was 9 and 6, respectively. There was a strong correlation between the number of FACS articles and the number of markers used per cancer type (R${}^{2}$ = 0.94, *p*-value = 0.005, Pearson correlation, Appendix A). Interestingly, we observed that FOXP3 was less frequently used to define Tregs in FACS studies of breast cancer, than in FACS studies of the other four types of cancer, being used in less than half the breast cancer articles. Finally, there was no major difference in Treg definition between blood and tissue Tregs (Figure 2c).

To conclude on the phenotypic definition of Tregs in cancer studies, there was a clear consensus in favor of the use of FOXP3 in IHC studies, but the situation was much less clear for FACS studies. FOXP3 was widely used, together with the combinations CD4/CD25 or CD4/CD25/FOXP3, with the frequent use of a threshold for the expression of CD25 or CD127.

### 3.2. The Contribution of Tregs to Prognosis Depends on the Treg Population

The lack of a standardized combination of markers to define Tregs makes it more difficult to draw conclusions about the role of these cells in cancer and their prognostic impact. A second problem is that some studies considered only specific regulatory subpopulations. We therefore decided to evaluate the contribution of studies on Treg subsets (Figure 3a) to improve the consensus on Tregs prognostic role, stratifying the data for cancer type. We first determined whether we could improve the consensus on the contribution of Tregs to prognosis by considering the information provided by the Treg population, for each type of cancer, separately. We then tried to determine whether a higher granularity could provide more reproducible conclusions on the link between Tregs and cancer across cancer types.

The number of subset studies available differed between types of cancer. For ovarian cancer, we were unable to identify any articles focusing on subsets (Figure 3b). We calculated one minus the normalized Shannon entropy and Fleiss’ kappa to evaluate the consensus, considering the effect of subset studies on prognosis: the higher the value obtained, the stronger the consensus between the different articles evaluated. We used a three-step approach: (i) we considered all studies simultaneously for a single cancer type; (ii) we considered only studies claiming to investigate the whole Treg population and (iii) we calculated Shannon entropy and Fleiss’ kappa separately for each Treg subset and calculated the weighted mean, with the number of patients used for weighting. For the calculation of this mean, the whole Treg population was counted as a subset. The coefficient of agreement ranges between 0 and 1, and is close to 1 if there is a consensus, or close to 0 if there is not. Our results showed no clear trend for all studies considered together and for studies dealing with the whole Treg population. However, the degree of consensus between studies was clearly increased by considering the mean of the entropies for each subset (Figure 3c, Appendix A): this implies that focusing on particular Treg subsets might improve the link between Tregs and cancer prognosis.

Strikingly, none of the articles focusing on Treg subsets reported a neutral role for Tregs in cancer prognosis. This provides a strong argument in favor of studying regulatory subsets in the TIME, particularly for translational applications. Furthermore, all regulatory subsets with an activated or similar phenotype, and the resting population (Figure 3a) were negatively related to prognosis, regardless of cancer type (Figure 3d). This negative relationship was observed even in colorectal cancer, for which the consensual claim is that Tregs have a positive impact on clinical outcome (Figure 3d). However, subset studies accounted for only a small fraction of the total cohort of patients for each cancer type (<1%, 2%, 7%, 23% for breast, colorectal, gastric and lung cancer, respectively). Conversely, the terminally activated regulatory fraction (Figure 3a) was associated with a good prognosis, but was studied in only one publication on lung cancer (Figure 3d). In this meta-analysis, we also explored the effect of the widely used CD25^high^ marker, considering: (i) articles using CD25^high^ (n = 6); (ii) articles based on simple positivity for CD25 (n = 21); and (iii) articles not using CD25 at all (n = 49). The rationale behind this exploration comes from the hypothesis that CD25^high^ is a reliable marker of regulatory cells, as it eliminates contaminating activated CD4 helper cells [36]. Five of the six articles using the CD25^high^ marker reported a negative association between Tregs and cancer prognosis (Figure 3e). The only article reporting a positive link considered terminally activated Tregs from the blood [37], while is has already been identified above as a subpopulation associated with a good prognosis (Figure 3d). The consensus for the CD25${}^{+}$ population was only slightly stronger (1− Shannon entropy = 0.166) than that for Tregs not delineated with CD25 (1− Shannon entropy = 0.164), and was much weaker than that for the CD25^high^ fraction (1− Shannon entropy = 0.35) (Appendix A). These findings strongly suggest that the CD25^high^ fraction is the one of interest, and is reproducibly associated with a poor cancer prognosis.

### 3.3. The Prognostic Value of Tregs Is Context-Dependent, as Shown by Analyses of Ratios to Other Cells in the Local Environment

Several articles investigated the correlation between Tregs and other cell populations from the same environment. This approach was used in 36% of the articles on breast cancer, 23% of those on colorectal cancer, 34% of those on gastric cancer, 19% of those on lung cancer and 29% of those on ovarian cancer. CD8${}^{+}$ T cells were always positively associated with Tregs, regardless of cancer type, and most articles looking at neighboring cells measured this association. The same positive correlation was found for tumor cells, CD3${}^{+}$ cells, cancer-associated fibroblasts (CAFs), follicular helper T cells (Tfh) and predendritic cells (DCs), across cancer types. Positive correlations with other cell types, such as myeloid cells, including myeloid-derived suppressor cells (MDSCs), macrophages and tumor-associated macrophages (TAMs), were described in just one article on breast cancer and one on ovarian cancer. Natural killer cells (NKs) and Th17 cells were negatively correlated with Tregs in one and three lung cancer studies respectively, as well as FOXP3${}^{+}$ tumor cells in one colorectal cancer article (Figure 4a).

We explored the role of neighboring cells in the impact of Tregs on prognosis, using the methodology described above, with the information about the use of an absolute Treg quantification or a ratio of Tregs to another cell population. Ratio-based studies accounted for 25% of the total cohort, but with different weights for each cancer type: 26% for breast cancer, 6% for colorectal cancer, 21% for gastric cancer, 11% for lung cancer and 41% for ovarian cancer. Again, a better consensus was obtained with ratios than with the absolute quantification (Figure 4b). This finding is logical, because ratios would partially take into account other components of the local environment, thereby better representing the complexity of the local environment. Some ratios were systematically correlated with a poor prognosis (Treg/CD8${}^{+}$ T cells), whereas others were systematically correlated with a good prognosis (Treg/Th17 cells). No trend was observed for the following ratios: Treg/CD4${}^{+}$ T cells or Treg/T cells (Appendix A).

### 3.4. Tumor Tissue Tregs Have a Greater Impact on Prognosis Than Blood Tregs

Finally, we also studied the role of Tregs detected in the patients’ peripheral blood or directly in the diseased organ, and even in specific parts of the tumor. There was no standardized way to name the different parts of the tumor. We therefore merged the various denominations used by the authors: intra-epithelial (or nest) vs. stroma, intratumoral vs. peritumoral. The “nest” was defined as the cells surrounded by cancer cells, whereas the “stroma” designated cells from the tumor stroma, i.e., patches of cells almost free of cancer cells within the tumor. “Peritumoral” was used to describe cells at the margin, whereas “intratumoral” was used to describe cells at the center of the tumor. We assessed the degree of consensus, as described in the Section 2, and we observed that the agreement on prognosis for each individual anatomic site was better than that for all anatomic sites together or for the non-segmented piece of tumor (Figure 5a). Overall, we found that anatomic site was a crucial parameter to understand the role of the Treg population in each cancer type, because entropy increased strongly when we took anatomic site into account in our evaluation of prognostic value (Figure 5a, Appendix A). In particular, for lung cancer, the strongest consensus was obtained only when the whole piece of tumor was taken into account. This reflects the higher degree of discrepancy in prognosis assessment based on blood Tregs, with most lung cancer studies being based on blood Treg detection. We suspect that the particularly high level of ambiguity for blood results is due to the Tregs from blood samples being identified by FACS, for which there is less consensus about the most appropriate markers than for IHC. We also suspect that blood samples do not reflect the TIME as well as tissue samples, therefore providing a less accurate representation of the tumor context [38,39].

For breast cancer, tissue Tregs were indicative of a good prognosis only for triple-negative breast cancers, and were reported to have a neutral effect in a single study with a very small cohort of patients (n = 40) [40], or if peritumoral Tregs were considered [41]. For the other cancer types, interpretation was less straightforward, as Tregs from different parts of the tissue also gave rise to different conclusions (Figure 5b). For colorectal cancer, almost all the anatomic sites displayed both positive and negative associations of Tregs with cancer prognosis, depending on the article considered, except for the juxtatumoral site and blood Tregs. A similar result was obtained for tumoral and intratumoral Tregs in gastric cancer, and for blood and stromal Tregs in lung cancer (Figure 5b). However, all but one of the articles that concluded that Tregs were associated with a good prognosis for cancer used IHC, which could not distinguish between the different Treg fractions described above. Furthermore, only one of the 15 articles used a ratio-based quantification.

## 4. Discussion

The hypothesis underlying this study was that the heterogeneity of regulatory phenotypes and their context are key parameters at least partly explaining the apparent discrepancies in the impact of Tregs on cancer prognosis. This meta-analysis shows that explicit description of Tregs and their afferent context could help to improve our understanding of the clinical role of these cells. We considered three different factors that could interfere with the pathophysiological role of Tregs: the different regulatory populations studied when evaluating the prognosis, the quantification method used and the anatomic microenvironment. We considered 130 articles focusing on five different types of cancer: breast, colorectal, gastric, lung and ovarian cancers. The amount of information available differed considerably between parameters, making it impossible to compare their relative importance directly, but we found that considering these parameters separately increased the resolution of the link between Tregs and cancer prognosis, in most of the situations considered. This finding contrasts with other reviews and meta-analyses considering only tumor characteristics, such as site or stage, to stratify patients and investigate the association between regulatory cells and prognosis [10,11,20,42,43].

We were able to detect reproducible associations between Tregs and cancer prognosis in some particular cases. For Treg subsets, the activated subpopulation was systematically found to be associated with a poor prognosis, as previously reported [36]. Thus, the way in which Tregs are defined and their different subsets have an impact on their prognostic value, as they exhibit various regulatory capacities. Taking the heterogeneity of these cells into account in our study helped to clarify their role in the TIME.

However, the context of Tregs is another factor, in addition to heterogeneity, affecting their prognostic value. The importance of context is illustrated here by the cross-talk with neighboring cells. We found that the Treg/CD8${}^{+}$ T cell ratio was systematically associated with a poor prognosis, and that Tregs were inversely correlated with CD8${}^{+}$ T cells regardless of cancer type.

Interestingly, these conclusions about both the diversity and cellular context of Tregs provide a unified view of the role of Tregs across cancer types, as opposed to the classical conclusions of published reviews [11,21,44]. Our methodology could be applied to other cell types, such as Th2 or Th17 cells, with as yet poorly defined roles [11], albeit to a lesser extent than for Tregs.

Our meta-analysis also highlights the differences in the design of studies of Tregs in the context of cancer, making it difficult to draw any clear conclusions on the contribution of Tregs to cancer prognosis. In light of our results, we suggest the following guidelines for new studies of Tregs in a cancer context: (i) focus on the CD45RO${}^{+}$FOXP3^high^ activated and CD45RO${}^{-}$FOXP3^low^ resting subsets, and the CD25^high^ fraction, (ii) determine the Treg/CD8 ratio, (iii) choose the study sample carefully (nest vs. stroma or intra- vs. peritumoral), (iv) annotate clinical data comprehensively. By following these guidelines, scientists and clinicians should be able to develop a more plausible description of the clinical role of Tregs for any cancer type. The same parameters should also be carefully considered when analyzing published studies.

## 5. Conclusions

We included three parameters in our analysis. We left out other parameters defining alternative contexts, such as treatment, because too little information was available, but the analysis of such meta-data would also be of great interest [45,46]. In particular, we could look into the differences between patients treated or not with immunotherapy.

Furthermore, we mentioned the importance of testing Treg regulatory capacities, since the definition of these cells is mainly functional. However, we could not include this element in our analysis, despite its relevancy, since the regulatory capacity quantification was not performed in a comparable manner across the different articles included in the study. In fact, only a small number of studies performed functional assays (27%, 61%, 26%, 38% and 32% for breast, colorectal, gastric, lung and ovarian cancers respectively).

Another line of investigation would involve the cross-analysis of all the available information about treatment, cancer subtype, anatomic site, definition of the global regulatory population and its subsets, and Treg quantification, to provide a clearer picture of the role of Tregs in the TIME. An exciting way to address these questions could be to use omics methods to collect and cross-analyze even more information, about the inflammatory context, for example. The advent of omics technologies has raised hopes for the identification of core gene expression signatures delineating the regulatory population. However, this has proved difficult, because these signatures remain dependent on the strategy used to capture the population of interest in the first place. Three articles identified signatures of 294, 136 and 31 genes, but the intersection between these signatures contains only 10 genes, including FOXP3, and CTLA-4, but not CD25 (IL2RA), which was not in the signature described by [47] despite a capture strategy using CD25^high^ (Appendix A).

We believe that this work may have implications not only at the bench, but also at the patient’s bedside. We found that activated regulatory phenotypes (FOXP3${}^{+}$CD45RO${}^{+}$ regulatory cells) were systematically associated with a poor prognosis. These cells may therefore constitute a key regulatory population to target and eliminate in the TIME, while sparing other Treg subsets. Additionally, it suggests that CD45RO is an important marker of the regulatory function. Similarly, boosting CD8${}^{+}$ T cells to disrupt and decrease the Treg/CD8${}^{+}$ T cell ratio might also improve patient outcomes, independently of the Treg diversity.

## Figures and Tables

**Figure 1 cancers-14-02991-f001:**
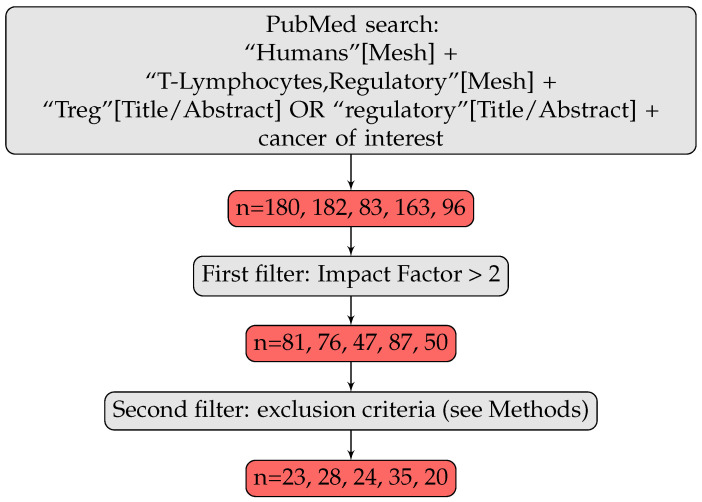
Article selection strategy. n is the number of articles retained at each step for the various types of cancer considered, listed in the following order: breast, colorectal, gastric, lung and ovarian cancers.

**Figure 2 cancers-14-02991-f002:**
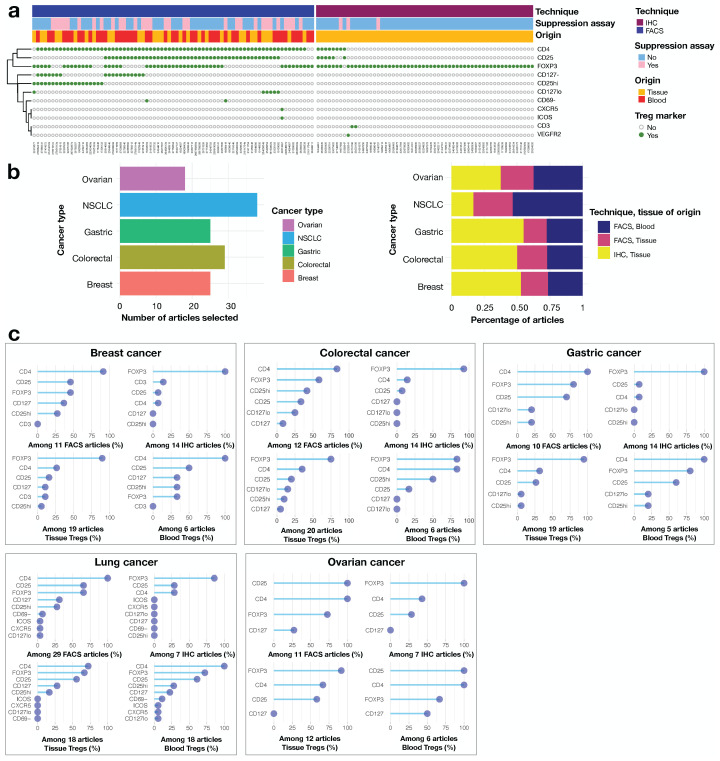
Treg markers used to identify Tregs in published articles on cancer. (**a**) The heatmap provides a clearer picture of the markers and combinations of markers most widely used to define human Tregs, based on 112 published articles on cancer. Each row corresponds to a marker and each column corresponds to an article. The heatmap on the left summarizes the markers used for FACS, together with a clustering of markers to represent common co-occurrences. The heatmap on the right summarizes the markers used for immunohistochemistry (IHC). It also indicates whether the suppression capacities of Tregs were tested in a functional assay, and the tissue from which the Tregs were obtained. (**b**) Histogram of number of articles included per cancer type (left), normalized histogram per cancer type, color-coded according to the technique used for Treg detection and the tissue of origin (right). (**c**) Lollipop graphs depicting the frequency of use of each Treg marker, by cancer type, technique for Treg detection and tissue of origin.

**Figure 3 cancers-14-02991-f003:**
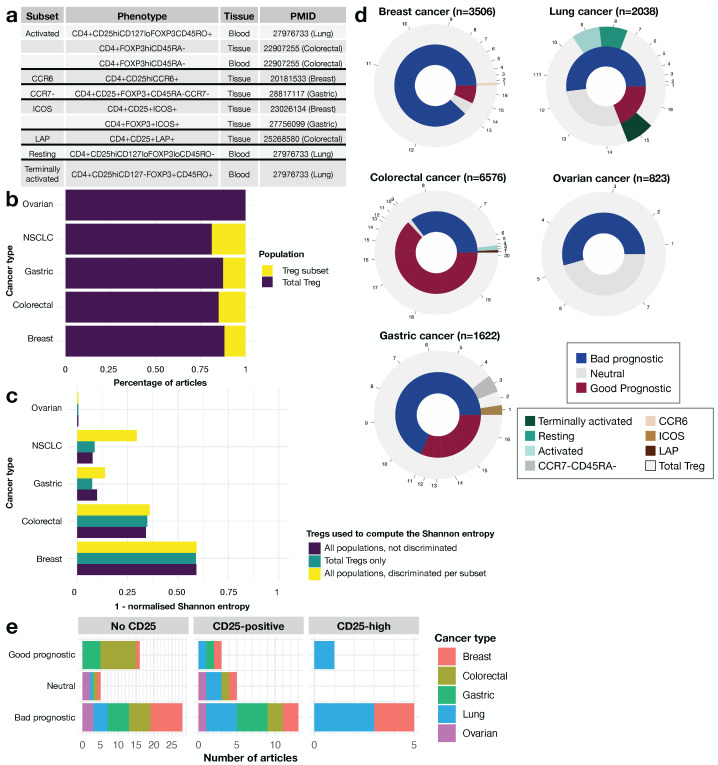
Subset diversity in cancer Tregs. (**a**) Summary of the various Treg subsets studied in the articles included in this meta-analysis. (**b**) Frequency of articles studying either the whole Treg population or a specific subset. (**c**) One minus the normalized Shannon entropy for each of the five cancer types, for all articles together (dark blue), articles focusing on the whole Treg population only (dark green), and mean of the entropies for each population type (yellow). (**d**) Pie chart of the prognostic impact of Tregs, as a function of the type of population used in the analysis, for each cancer type. Each numbered portion is an article and its size reflects the number of patients included in the study. (**e**) Bar plot of the prognostic value of Tregs in articles using CD25^high^ (right), CD25${}^{+}$ (middle), or no CD25 (left) to delineate Tregs.

**Figure 4 cancers-14-02991-f004:**
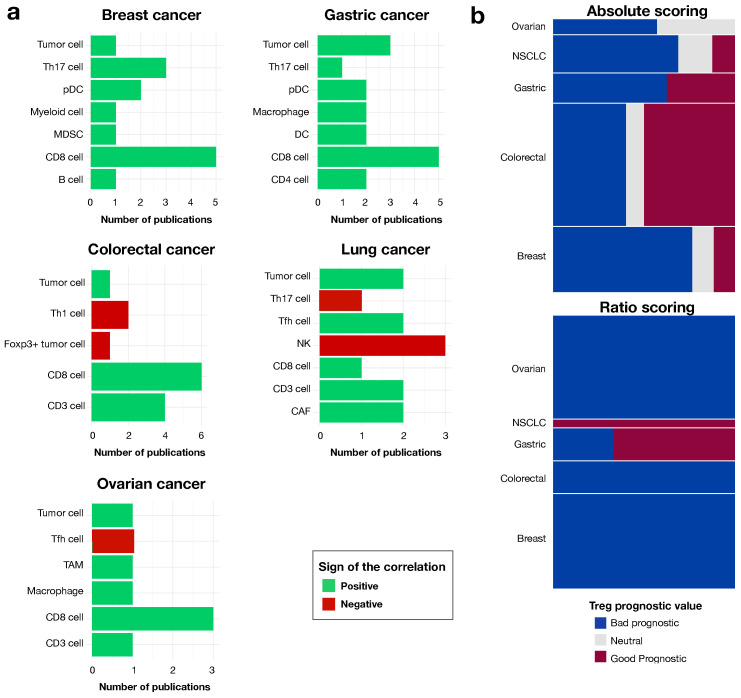
Interplay between Tregs and neighboring cells. (**a**) Correlation between Tregs and other cell populations, by cancer type; number of articles depicting the various correlations. (**b**) Treemap of the prognostic value of Tregs by type of quantification: absolute quantification (top panel) or quantification via the determination of a ratio (considering all Treg/neighboring cell ratios) (bottom panel). The length of each bar represents the proportion of patients for each prognostic value by cancer type, and the height of each rectangle indicates the proportion of patients from each cancer type (n = 14,565 for absolute quantification, n = 3653 for ratio quantification).

**Figure 5 cancers-14-02991-f005:**
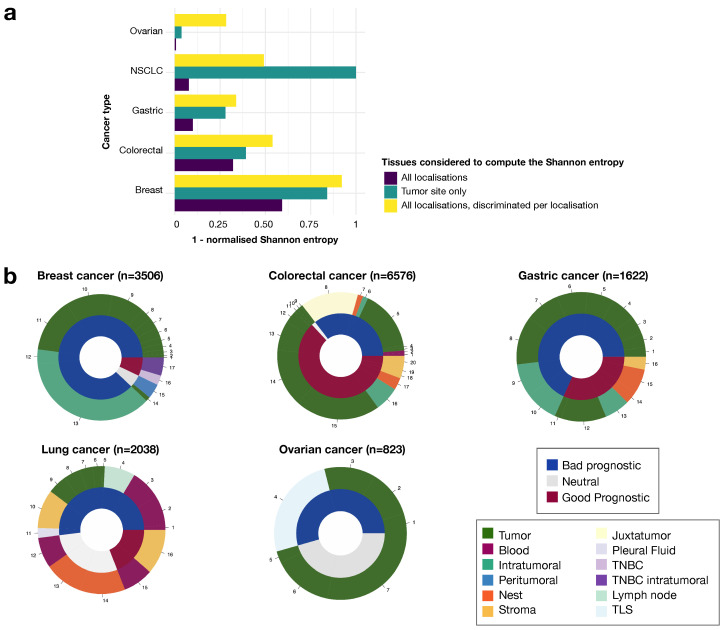
Interplay between Tregs, their anatomic locations and their prognostic value. (**a**) Histogram showing one minus the normalized Shannon entropy for each cancer type and for each group, by anatomic site: all locations together (dark blue), tumor site (dark green), and mean of the entropies for each anatomic site (yellow). (**b**) Pie charts of prognostic value by anatomic site, for each cancer type. Each numbered portion is an article and its size reflects the number of patients included in the study. (TNBC: triple-negative breast cancer; TLS: tertiary lymphoid structure).

## Data Availability

Refer to Appendix A.

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
