# Peer review of "Context-Dependent Effects Explain Divergent Prognostic Roles of Tregs in Cancer"

_cancers, 2022, doi:10.3390/cancers14122991_

Round 1
Reviewer 1 Report
I think the topic is very relevant. The selection of markers for the identification of Treg is a very complex issue. There is no general agreement on the issue and the choice of the researcher varies a lot. Nobody knows what specific marker one should look for.
The article is a meta-analysis of already published research in that sense it is original. However, the issues have been discussed earlier.
I found it easy to understand.
The discussion is much more futuristic and sometimes feels irrelevant to the current manuscript.
The research is more descriptive rather than testing some hypotheses. In that says, they just described the issue the field is facing. I found the conclusion of the paper is relatively weak. The author should have come up with some solid suggestions/guidelines regarding what marker to use/not to use. However, I understand that it will be hard for them to recommend something. Usually, such recommendations are provided by a network of researchers.· What is the color code in Fig 1 b. Please specify them.
Author Response
I think the topic is very relevant. The selection of markers for the identification of Treg is a very complex issue. There is no general agreement on the issue and the choice of the researcher varies a lot. Nobody knows what specific marker one should look for.
The article is a meta-analysis of already published research in that sense it is original. However, the issues have been discussed earlier.
I found it easy to understand.
The discussion is much more futuristic and sometimes feels irrelevant to the current manuscript.
The research is more descriptive rather than testing some hypotheses. In that says, they just described the issue the field is facing. I found the conclusion of the paper is relatively weak. The author should have come up with some solid suggestions/guidelines regarding what marker to use/not to use. However, I understand that it will be hard for them to recommend something. Usually, such recommendations are provided by a network of researchers.
Reply 1: We thank the reviewer for these comments. We tried to improve the conclusion according to their remarks, but it is indeed not realistic to suggest the best markers to use simply from our results. We could only suggest markers for Tregs in the context of human cancer studies (lines 388-389).
What is the color code in Fig 1b. Please specify them.
Reply 2: We thank the reviewer for pointing this omission out. Color code has been added in the first panel of Figure 2b.
Reviewer 2 Report
The authors here describe with an elegant meta-analysis the prognostic role of Tregs in cancer. In addition, they establish guidelines for improving the design in future studies to evaluate Tregs in cancer. This is an increasingly relevant topic in the tumor immunology field, and therefore highly interesting for the readership of cancers. The authors manage to discuss this under-evaluated research topic in a comprehensive, critical, and clear manner.
Comment 1: The authors excluded studies reporting on the prognostic role of Tregs in patients receiving immunotherapy. Especially since the field of immunotherapy is growing rapidly (as noticed by the authors themselves in the second sentence of this manuscript), I believe the authors should strongly consider to include these patients in their analysis, or a sub-analysis as this would highly add the impact of the paper. Else, the authors need to clearly describe why they exclude these patients from their meta-analysis.
Comment 2: It is unclear from the manuscript when Treg values are reported by the studies. It is highly relevant whether this is prior to, during, or after treatment. The authors should report on this, and consider this in their analysis. Also, it would be valuable to know the treatments applied in the studies, and if this matters for the prognostic value of Tregs.
Comment 3: Although the authors briefly touch upon the role of testing Treg function in Figure 2, the issue of Treg function in light of prognostic value is not discussed at all. Since there is increasing evidence that T cell function (rather than quantity only) is relevant for patient outcome, this should touched upon in the discussion.
Comment 4: In line 292, the authors describe that: “We also suspect that blood samples do not reflect the TIME as well as tissue samples, therefore providing a less accurate representation of the tumor context.” Are there references that can back up this suspicion? Comparing the level of Tregs in the tumor and the blood?
Author Response
The authors here describe with an elegant meta-analysis the prognostic role of Tregs in cancer. In addition, they establish guidelines for improving the design in future studies to evaluate Tregs in cancer. This is an increasingly relevant topic in the tumor immunology field, and therefore highly interesting for the readership of cancers. The authors manage to discuss this under-evaluated research topic in a comprehensive, critical, and clear manner.
Comment 1: The authors excluded studies reporting on the prognostic role of Tregs in patients receiving immunotherapy. Especially since the field of immunotherapy is growing rapidly (as noticed by the authors themselves in the second sentence of this manuscript), I believe the authors should strongly consider to include these patients in their analysis, or a sub-analysis as this would highly add the impact of the paper. Else, the authors need to clearly describe why they exclude these patients from their meta-analysis.
Reply 1: We decided to exclude those patients because immunotherapy will disturb the tumor immune micro-environment while we are interested in the prognostic value of untreated Tregs. We modified the manuscript to explain this choice (lines 101-103).
Comment 2: It is unclear from the manuscript when Treg values are reported by the studies. It is highly relevant whether this is prior to, during, or after treatment. The authors should report on this, and consider this in their analysis. Also, it would be valuable to know the treatments applied in the studies, and if this matters for the prognostic value of Tregs.
Reply 2: We understand this relevant comment about the timing of the treatment. However, treatment information is again too scarce : we did not find information at all regarding the treatment regimen for 19% of breast, 45% of colorectal, 14% of gastric, 12% of lung and 42% of ovarian cancer articles. We reported whenever available Treg values prior to treatment, otherwise it is after treatment. We modified the methods section to add this information (lines 103-106). Additionally, 50% of breast cancer studies included in our analysis enrolled patients undergoing treatment, but it concerns only 32% of colorectal, 2% of gastric, 8% of lung and 11% of ovarian cancer papers. Unfortunately we did not have enough information about treatment to include it in the analysis (chemotherapy or radiotherapy regimen, surgery etc). We already discussed in the original version of the manuscript that it would be however of great interest (lines 363-364).
Comment 3: Although the authors briefly touch upon the role of testing Treg function in Figure 2, the issue of Treg function in light of prognostic value is not discussed at all. Since there is increasing evidence that T cell function (rather than quantity only) is relevant for patient outcome, this should touched upon in the discussion.
Reply 3: We thank the reviewer for this interesting comment. In fact, we have only part of the information as the quantification of the regulatory efficiency was made in a very heterogenous way in the different articles we screened. In the papers included in our study, Tregs would be either untested (we coded it as "0" status in an attempt to standardize notations), or termed as suppressive ("1" status) or highly suppressive ("2" status). The repartition of statutes 0/1/2 is the following (in %): 73/27/0, 39/45/16, 74/22/4, 62/38/0 and 68/22/10, for breast, colorectal, gastric, lung and ovarian cancer papers respectively. Hence, we do not have enough information about the functional assays to include it in our study. However, we completely agree that the functional status matters, and we have modified the conclusion accordingly to mention this fact (lines 366-371).
Comment 4: In line 292, the authors describe that: “We also suspect that blood samples do not reflect the TIME as well as tissue samples, therefore providing a less accurate representation of the tumor context.” Are there references that can back up this suspicion? Comparing the level of Tregs in the tumor and the blood?
Reply 4: We thank the reviewer for this interesting suggestion. We should indeed back up our intuition with existing experimental proofs. We have now included two relevant references in the revised manuscript (line 305, references 38 and 39).
Reviewer 3 Report
This is an interesting study on the prognostic role of Treg in certain types of cancers, which could guide the direction of the future studies on this topic. The authors may want to address the following points. 1_ Title: Please revise the title so that the readers recognize that this study is a meta-analysis on BrCa, Lung ca, etc. 2_ M&M: The authors need to clarify that they followed one of guidelines such as the PRISMA and may want to provide information on whether this study meets the requirements and the results of the quality assessment of studies included. 3_ Lines 94-5: It would be better to mention the year in which the authors referred to impact factors of journals publishing cited articles.Author Response
This is an interesting study on the prognostic role of Treg in certain types of cancers, which could guide the direction of the future studies on this topic. The authors may want to address the following points.
1_ Title: Please revise the title so that the readers recognize that this study is a meta-analysis on BrCa, Lung ca, etc.
Reply 1: We thank the reviewer for mentioning that. However, we deemed it more meaningful to not mention in the title directly the type of cancers that we selected, since this study is intended to assess the prognostic role of Tregs in cancer in general. We only chose these 5 cancer types to span the whole spectrum of Treg prognostic values.
2_ M&M: The authors need to clarify that they followed one of guidelines such as the PRISMA and may want to provide information on whether this study meets the requirements and the results of the quality assessment of studies included.
Reply 2: We did not follow the PRISMA guideline from the beginning as we were not aware of it, but we checked a posteriori that we were meeting its requirements for relevant items among their checklist (http://prisma-statement.org/PRISMAStatement/Checklist). We modified the methods section accordingly to mention the items that were consistent with PRISMA guidelines (lines 110-113).
3_ Lines 94-5: It would be better to mention the year in which the authors referred to impact factors of journals publishing cited articles.
Reply 3: We thank the reviewer for pointing out this omission. It was the 2-year impact factor of 2019. We modified the text accordingly (line 95).
Round 2
Reviewer 2 Report
All my comments were sufficiently answered by the authors.
Reviewer 3 Report
The authors addressed points raised by the reviewer.